# Simulated Swine Digestion and Gut Microbiota Fermentation of Hydrolyzed Copra Meal

**DOI:** 10.3390/ani14111677

**Published:** 2024-06-04

**Authors:** Jurairat Rungruangsaphakun, Francis Ayimbila, Massalin Nakphaichit, Suttipun Keawsompong

**Affiliations:** 1Specialized Research Unit: Prebiotics and Probiotics for Health, Department of Biotechnology, Faculty of Agro-Industry, Kasetsart University (CASAF, NRU-KU), Bangkok 10900, Thailand; jurairat.run@ku.th (J.R.); ayimbilalike1@gmail.com (F.A.); fagimln@ku.ac.th (M.N.); 2Center for Advanced Studies for Agriculture and Food, KU Institute of Advanced Studies, Kasetsart University (CASAF, NRU-KU), Bangkok 10900, Thailand

**Keywords:** hydrolyzed copra meal, swine gut microbiota, swine feces, mannanase, weaned swine, SCFA

## Abstract

**Simple Summary:**

This study investigated the impact of adding 1% processed coconut meal (hydrolyzed copra meal, HCM) to the diets of weaned pigs, comparing it to regular coconut meal (CM) and a control diet with no coconut meal. Processed to enhance digestibility, HCM showed better digestion than CM. Both treatments supported a diverse gut microbiome, with potential benefits for producing beneficial fatty acids crucial for gut health. More research is needed to fully evaluate HCM’s impact on pig growth and health, but initial findings suggest that incorporating HCM could enhance digestive health and overall well-being in young pigs.

**Abstract:**

This study aimed to compare the effects of hydrolyzed copra meal (HCM) inclusion at 1% on its in vitro digestibility and the microbiota and cecum fermentation using the gut microbiota of weaned swine, targeting microbial community and short-chain fatty acids (SCF). For this reason, three treatments were considered: control (no copra meal), 1% non-hydrolyzed copra meal (CM), and 1% HCM. Non-defatted copra meal was hydrolyzed and analyzed (reducing sugars and total carbohydrates) in our laboratory. For digestion, microbiota identification, and fermentation assays, fresh fecal samples from two weaned pigs (1 month old) were used. Three replicates of each treatment were employed. HCM was more digestible, with approximately 0.68 g of hydrolysate recovered after simulated digestion compared to 0.82 g of hydrolysate recovered from CM. This was shown by Scanning Electron Microscope (SEM) images. Also, the three swine shared the majority of microbial species identified at the phylum and family levels. There were no differences (*p* > 0.05) between treatments in the microbial community and SCFA during fermentation. However, higher Chao-1 and Shannon indexes were observed in CM and HCM treatments. HCM was also found to be capable of preserving *Actinobacterota* and *Proteobacteria* at the phylum level, while at the family level, both treatments may help *Lactobacillaceae*, *Peptostreptococcaceae*, *Lachnospiraceae*, and *Ruminococcaceae* survive in the long term. Also, there was a potential trend of increasing acetic acid and butyric acid in the CM and HCM treatments. While HCM shows promise in potentially modulating the gut microbiota of weaned swine, additional research is required to investigate the effects of higher doses of HCM on swine performance parameters.

## 1. Introduction

Mannooligosaccharide (MOS) provides a variety of health benefits, which tailor its applications in the animal industries [1]. MOS derived from plants can modulate the gut microbiota by enhancing protection in the intestinal mucosal layer and boosting immunity, antimutagenic, and antioxidant defenses [2]. However, the application of MOS is limited to animals and lacks commercialization, when compared to yeast cell walls [1,3]. This is a result of a lack of systematic research on the mechanisms underlying their biological benefits. MOS has been synthesized from copra meal (CM) using an enzyme, which produces mostly β-anomers and some α-anomers [4,5]. This type of MOS may have health benefits similar to yeast MOS, which only contains α-anomers. MOS with β-anomers is more likely to escape digestion and become available to the swine gut flora [1].

Copra meal (CM) has evolved into a valuable animal feed over the years, becoming even more efficient when pretreated [6]. Hydrolyzed copra meal (HCM) is a newly produced functional ingredient by β-1,4-mannanase hydrolysis of defatted or non-defatted copra meal. HCM has a reduced fiber content (NDF, ADF, cellulose, hemicellulose, and lignin) and increased soluble neutral detergent [5]. The fiber content, mainly galactomannan, is broken down into mannooligosaccharides (MOSs), mostly mannobiose, mannotriose, mannotetraose, mannopentaose, and mannohexaose, as well as reducing sugars such as mannose and glucose [5,7]. Thus, HCM may ensure several benefits to swine, including eliminating the antinutritional compounds present that impair the performance of swine. Also, the degraded carbohydrates and proteins readily absorbed can supply nutrients needed by swine for maintenance and production. In addition, the manno-oligosaccharides produced can function as prebiotics or modulate microbial ecology in the gut to improve the swine’s health, which is the focus of this study.

A balanced gut microbial ecology influences host health, which is affected by dietary supplements such as prebiotics [8]. The potential effects of HCM on the gut microbiota of animals have been documented. According to Prayoonthien et al. 2018, copra meal hydrolysate stimulated the growth of lactobacilli in a medium containing copra meal hydrolysate and ileum extract [9] H-DCM could enhance the growth of both *Lactobacillus reuteri* KUB-AC5 and *Lactobacillus johnsonii* KUNN19-2 and reduced *Escherichia coli* KUB-E010 growth [10]. In another study, MOS produced from CM by mannanase hydrolysis under ideal conditions increased the growth of Lactobacilli and Bifidobacteria but inhibited the development of enteropathogenic *Salmonella* enteritidis S003, *E. coli* E010, *Staphylococcus aureus* TISTR 029, and *Shigella dysenteriae* DMST 1511 [7]. This demonstrates that some information about HCM’s ability to boost beneficial bacteria and inhibit pathogens is now available. However, HCM behavior in the swine gut microbiota in terms of SCFA and the modulation of gut flora is limited. Furthermore, the discovery of metabolome signatures in suckling and weaned piglets opens the door to the development of health-promoting nutritional strategies that promote swine health [11]. Therefore, this study evaluated the colonic fermentation of HCM using the gut microbiota of weaned swine, targeting the microbial community and short-chain fatty acids. Before introducing sample treatments, a comparison of the intestinal microbial structure between suckling and weaning swine was conducted to explore the potential effects of environmental conditions on the gut microbiota of swine.

## 2. Materials and Methods

### 2.1. Materials

Copra meal was obtained from the coconut milk industry, Ampol Food Processing Co., Ltd., Nakhon Pathom, Thailand, dried, and milled using a hammer mill to obtain a particle size of 0.5 mm. For enzyme production, the defatted copra meal was prepared by oil extraction in a Soxhlet apparatus for 4–6 h. For hydrolysis, dried typical copra meal (non-defatted) from the Thai coconut milk industry was used. *Bacillus circulans* NT 6.7 was obtained from the Department of Biotechnology, Kasetsart University [12]. Pepsin from the porcine gastric mucosa (P7000, Sigma-Aldrich, St. Louis, MO, USA), Viscozyme (V2010, Viscozyme L, Sigma-Aldrich, USA), and pancreatin from the porcine pancreas (P1750, Sigma-Aldrich, USA) were employed.

### 2.2. Hydrolysis Process of Copra Meal and Reducing Sugars and Total Carbohydrates Determination

The hydrolysis of copra meal (CM) was performed to achieve hydrolyzed copra meal (CMH) using mannanase from *Bacillus circulans* NT 6.7 according to the previous fed-batch method [5]. This was conducted in a 1 L reactor containing 500 mL of 18 U/mL mannanases (pH 6.0), a substrate concentration of 25 g, a stirring speed of 250 rpm, and a start time of 1 h, for a total of 3 times for 6 h at 50 °C. The release of reducing sugars (RSs) in 5 min at 100 °C was measured as mannose equivalents using dinitro salicylic acid (DNS) [13]. Briefly, 100 µL of the sample and 100 µL of DNS were added into an Eppendorf tube, closed, covered for 5 min at 100 °C, and then immediately immersed in cold water. Next, the Eppendorf tubes were cooled and then 1 mL of distilled water was added to each before measurement for light absorption at 540 nm. The method for phenol–sulfuric acid [14] as a simple and rapid colorimetric method was used to determine total carbohydrates (TC) in the sample. Briefly, 500 µL of the sample and 500 µL of phenol were added into a test tube, mixed, added with 2.5 mL of sulfuric acid, and then mixed and cooled at room temperature. The absorbance was measured at 490 nm.

### 2.3. Digestion Evaluation of CM and HCM by Swine Simulated Gastrointestinal (GI) Conditions

The simulated GI system included the stomach and small intestine. Initially, 1 g of CM or HCM was combined with 50 mL of 0.1 M phosphate-buffered solution, pH 6.0, and the pH was changed to 2.0 with 1 mL of pepsin solution (50 mg/mL; 500 units/mg solid), incubated for 2 h at 39 °C with shaking. Following that, 20 mL of 0.2 M phosphate-buffered solution at pH 6.8 and 10 mL of 0.6 M NaOH solution were mixed to simulate the small intestine condition. After adjusting the pH to 6.8, 2 mL of pancreatin solution (100 mg/mL; 4USP, P1750) was added and incubated at 39 °C for 4 h while shaking [15]. The reducing sugars and total sugars were determined using the methods outlined above. Oligosaccharide release was examined using a Waters high-pressure liquid chromatography (HPLC) system with a refractive index detector (2414 RID; Waters, Milford, MA, USA) and the breeze 2 program. The operating temperature for the column was 75 °C and that for the detector was 40 °C, the mobile phase was 0.4 mL/min, and the injection volume was 20 µL. The device was calibrated using mannose, mannobiose (M2), mannotriose (M3), mannotetraose (M4), mannopentaose (M5), and mannohexaose (M6) (Megazyme, Ireland).

### 2.4. In Vitro Fermentation of Fecal Samples for Microbial Analysis and Volatile Fatty Acids Determination

Fresh fecal samples were taken from three swine from agricultural farms: one Suckling swine (1 week old) and two weaned swine (1 month old). Samples were stored in anaerobic culture swabs and used within 1 h. The DNA analysis of the microbiota in fresh feces from a suckling swine and weaned swine was evaluated and compared. This research was conducted according to the guidelines and the relevant regulations of the Kasetsart Announcement of the University Ethics Committee.

The fresh fecal samples taken from two weaned swine (1 month old) were used for fermentation in the presence of CM and HCM. Herein, three treatments were considered: control (no copra meal), 1% non-hydrolyzed copra meal (CM), and 1% HCM.

The fecal anaerobic fermentation system was prepared according to the method described by [16]. A 300 mL fermentation vessel (Lambda, Switzerland) and 150 mL of basal medium containing 2.0 g of Peptone, 2.0 g of yeast extract, 0.10 g of NaCl, 0.04 g of K_2_HPO_4_, 0.04 g of KH_2_PO_4_, 0.01 g of MgSO_4_.7H_2_O, 0.01 g of CaCl_2_.2H_2_O, 2.0 g of NaHCO_3_, 2 mL of Tween 80, 0.05 g of Hemin, 10 µL of Vitamin K, 0.50 g of L-cysteine, 0.5 g of Bile salt, and 4 mL (0.05 g/L) of Resazurin [17], autoclaved at 121 °C for 15 min, were used. Before use, L-cysteine and Sodium Thioglycolate solutions were sterilized separately with a 0.22 m filter. In the fermentation vessel, a sterile medium was maintained under anaerobic conditions by sparking the vessels with O_2_-free N_2_ (15 mL/min) overnight. The temperature was held at 39 °C using a circulating water bath, and pH values were controlled between 6.7 and 6.9 using an automated pH controller which adds acid or alkali as required (0.5 M HCl and 0.5 M NaOH). After the fermentation, fecal samples were homogenized with sterilized 0.1 M phosphate-buffered saline, pH 6.8, to obtain 10% (*w*/*v*) suspension. Each vessel was inoculated with 1.4 mL (1% of the total volume) of a freshly prepared fecal slurry from each donor each time. The control treatment consisted of a basal medium with fecal matter from weaned swine 1. Sample treatment 1 included a basal medium with fecal matter from weaned swine 1, supplemented with 1% copra meal (CM). Sample treatment 2 used the same basal medium with fecal matter from weaned swine 1, but supplemented with 1% hydrolyzed copra meal (HCM). Repeat the experiment but used the feces of the weaned swine 2. The content in each vessel was then stirred at 39 °C with a pH between 6.65 and 6.95. Sparing the vessels with oxygen-free nitrogen gas at a rate of 15 mL/min was performed to maintain anaerobic conditions. Samples were collected at 0 and 12 h of fermentation for DNA extraction for next-generation sequencing (NGS) and short-chain fatty acid (SCFA) analysis.

#### 2.4.1. DNA Extraction for Next-Generational Sequencing (NGS)

Genomic DNA was extracted from each sample using a combination of bead-beating [18]. One milliliter of culture sample obtained from each vessel at each sampling time was centrifuged at 15,000× *g* for 5 min. The pellet was washed twice with 1 mL of filtered sterilized PBS (0.1 mol/L, pH 7.4). The pellet was added with 800 µL of solution CD1 and vortexed to mix well. The content was transferred to 2 mL Power-Bead Pro tubes and secured on an MP Biomedical FastPrep 24 Tissue Homogenizer run at 6.5 m/s for 3 min (QIAamp^®^ PowerFecal^®^ Pro DNA Kit (Qiagen, Germany)). The tubes were then centrifuged at 15,000× *g* for 1 min. Next, 550 µL of the supernatant was transferred to a clean 2 mL microcentrifuge tube, and 200 µL of solution CD2 was added, vortexed for 5 s, and centrifuged at 15,000× *g* for 1 min. After that, 600 µL of solution CD3 was added and vortexed for 5 s. The solution of 650 µL of the lysate was loaded onto an MB Spin Column and centrifuged at 15,000× *g* for 1 min (this step was repeated to ensure that all the lysate had passed through the MB Spin Column). Then, 500 µL of solution EA was added to the MB Spin Column and centrifuged at 15,000× *g* for 2 min. The supernatant was discarded and the MB Spin Column placed back onto the same 2 mL collection tube. This step was repeated until all samples were loaded. The MB Spin Column was carefully placed on a new 1.5 mL elution tube and dried at 70 °C for 2 min. Then, 50–100 µL of solution C6 was transferred to the center of a white membrane and was incubated at room temperature for 5 min. Following that, it was centrifuged at 15,000× *g* for 1 min, and the MB Spin Column was discarded. The DNA was now ready for downstream applications (store the DNA at −20 °C as solution C6 does not contain EDTA). The concentration of DNA was measured using a NanoDrop 2000 (NanoDrop Technologies, Wilmington, NC, USA). The DNA sample was checked for amplification by PCR and gel electrophoresis.

#### 2.4.2. Microbial Analysis

MiSeq sequencing was used to analyze the microbiota after running agarose gel electrophoresis and PCR to determine the integrity of the DNA [19]. The genomic DNA from each sample was used to amplify the V3–V4 region of the 16s rRNA gene by using barcoded-forward primer 341F (CCTAYGGGRBGCASCAG) and reverse primer 806R (GGACTACNNGGGTATCTAAT) and the Phusion^®^ High-Fidelity PCR Master Mix (New England Biolabs, Ipswich, MA, USA). Then, a specific index sequence was added. For the purification of PCR products for library construction to generate 2–250 bp paired-end reads, high-throughput sequencing was performed on the Illumina NovaSeq 6000 platform (Illumina, San Diego, CA, USA). Using the previous protocols [20], all reads were combined using USEARCH v11.0.667 [20]. In a nutshell, the search_PCR2 and fastx_truncate commands were used to remove the primer from the merged reads. Utilizing the UNOISE algorithm to denoise the processed pair of sequences, the amplicon sequence variants (ASVs) table was created using the UPARSE pipeline. The SINTAX algorithm was used to assign taxonomies [21] and the Ribosomal Database Project (RDP) training set v18. ASVs with no phylum label or that are singletons were excluded from the analysis. USEARCH v11.0.667 was used to calculate the alpha and beta diversity indices.

#### 2.4.3. Short-Chain Fatty Acid Analysis during Swine Microbiota Fermentation

The samples were centrifuged at 13,000× *g* for 5 min and the supernatants were filtered using a 0.22 µm filter unit (Millipore, Cork, Ireland). The column used a combination of size exclusion and ligand exchange mechanisms. Aminex HPX-87H (Bio-Rad, Hercules, CA, USA) was maintained at 50 °C. The mobile phase was 8 mM H_2_SO_4_, the flow rate was 0.6 mL/min, a UV detector was equipped (Waters 2489 UV–visible detector, Milford, MA, USA), a wavelength of 210 nm was used, and the breeze2 program was used for analysis. SCFAs were quantified using calibration curves for lactic acid, acetic acid, propionic acid, and butyric acid at concentrations of 10.375, 20.75, 41.5, 83, 166, and 332 mol/mL, respectively. Tartaric acid (Sigma-Aldrich, Gillingham, UK) was used as the internal standard at a final concentration of 0.013% [22].

### 2.5. Statistical Analysis

Statistical analyses were carried out with the help of the statistical software XLSTAT 2019. The Kruskal–Wallis test for multiple pairwise comparisons using Dunn’s procedure/two-tailed test was used to determine statistical significance among the three treatments, control, HCM, and CM, when the *p*-value for interaction was <0.05. The statistical model takes into account the effects of CM, hydrolyzed HCM, and a negative control. Each experiment was conducted in two replicates. The Shannon, Simpson, and Chao-1 indices were used to calculate community diversity.

## 3. Results and Discussion

### 3.1. Hydrolyzed Copra Meal

The hydrolysis of copra meal (CM) resulted in a hydrolyzed copra meal (HCM) with an improved content of reducing sugars and oligosaccharides. HCM was also found to be more digestible than CM after simulated digestion in the large intestine of swine [5]. To better understand the functionality of HCM in swine, HCM was compared to CM in simulated stomach and small intestine digestion conditions. Using Scanning Electron Microscopy (SEM) analysis, the effects of digestion were visualized between CM and HCM. Following digestion in the small intestine, breakages of connective structures with holes on the surfaces of both samples were found, demonstrating the porcine pancreatin’s hydrolysis effect in small intestine conditions (4 h), as opposed to samples at 0 h. HCM, however, displayed more obvious surface holes than CM (Figure 1). This can be attributed to the severity of the aggregate separation and linkage breakdown of HCM structures induced by the stomach and small intestine conditions of swine. The SEM results were in support of the dry weight of CM and HCM collected after digestion in the simulated gastrointestinal tract of swine, which was 0.82 g and 0.68 g, respectively, compared to the initial sample weight of 1 g before (0 h) hydrolysis (Table 1). Additionally, varying oligosaccharides between CM and HCM were detected after simulated GI digestion. HCM is composed of oligosaccharides ranging from mannobiose (M2) to mannohexaose (M6), while CM contains only mannohexaose (M6) after stomach digestion (Table 2). This shows a greater effect of stomach conditions, particularly the acid on the HCM compared to CM. Finally, subjected to small intestine conditions, mannobiose (M2), mannotriose (M3), and mannotetraose (M4) were observed in HCM, while mannotriose (M3), mannopentaose (M5), and mannohexaose (M6) were contained in CM. It can be deduced that HCM had a lower degree of polymerization oligosaccharides (M2-M4) than CM in this stage of digestion, which can be attributed to HCM’s greater susceptibility to pancreatin enzymes. The low total oligosaccharide content can be attributed to the fact that most oligosaccharides observed in the stomach were converted to monosaccharides in the small intestine. The indication that HCM may be more digestible in swine GI results in more effective nutrient availability, which concerns a previous report [5]. CM and HCM were then subjected to in vitro colonic fermentation using swine feces, as shown in the subsequent section.

### 3.2. Understanding Microbiota of Swine Feces

The structure and function of the digestive tract are different in different physiological stages of animals. It was indicated that for growing–finishing pigs, which have strong immune systems and better health status, adding oligosaccharide to their diets will not show many beneficial effects [23]. It is easier for weaned piglets with a developing immune system and unstable intestinal microflora to get sick, and the main reason for diarrhea is the increase in pathogenic bacteria, so adding functional oligosaccharides after weaning can achieve a better effect [23]. Weaning is one of the most stressful events in swine farming. Swine are subjected to physiological, social, environmental, and nutritional stresses. Understanding the composition of the microbial community and its functional capacity during suckling and weaning is vital for swine production because bacteria play important roles in the swine’s health and growth performance [24]. Also, the identification of metabolome signatures in suckling and weaned piglets paves the way for the development of health-promoting nutritional strategies for swine health [11]. Thus, we first compared the microbial structure among suckling swine less than 2 weeks old, and two weaning pigs 4 weeks old.

As shown in Figure 2, Chao-1, Simpson, and Shannon indexes showed no significant difference (*p* > 0.05) among the three treatments. However, weaned pig 1 appeared to be the richest, followed by suckling pigs. The Chao-1 index indicated that the species richness in weaned pig 2 could be comparable to those of suckling pigs. The Simpson index indicates the evenness and diversity of the gut microbiota community and also showed significant variation (*p* > 0.05) among the swine. This indicated that microbial richness and diversity in young swine are influenced not only by age or developmental stage but also by how they respond to environmental conditions on the farm. The authors in [25] found that the bacterial richness and diversity of cecal microbiota increased in piglets after weaning and that there was a continuous increase in the alpha diversity of gut microbiota during the weaning transition. Other studies, however, reported a decrease in alpha diversity early after weaning and an increase later from weaning to adulthood [24]. This implies that the richness and diversity of gut microbiota in young swine may be influenced by a variety of factors, including age and farm management conditions such as feeding. In addition, bar graphs were used to illustrate the number of common and unique species in suckling pigs and weaned pigs. As shown in Figure 3 and Figure 4, the microbiota of the weaned and suckling swine shares the majority of the microbial species identified at the phylum and family levels with some unique bacterial abundance. No significant variation (*p* > 0.05) in the abundance was observed among the three swine. The disparities between studies conducted at various times may be the cause of the different results found between the current study and previous studies. These include differences in management and feeding practices and the microbiological environment of the farm. The stress that animals face on a farm can vary greatly, even between commercial farms. Weaning conditions may vary depending on farm size, environmental conditions, and sanitation conditions. Weaned pigs, though born on the same day, may have different microbiota as a result of different reactions to environmental stress and the amount of feed consumed.

The durations of digestion: stomach; 2 h, small intestine; 4 h.

### 3.3. Understanding the Modulation Effect of CM and HCM

#### 3.3.1. The Gut Microbiota Composition and Structure

In recent years, many studies have been reported on the application of functional oligosaccharides in swine breeding, especially in weaned swine. Oligosaccharides play a significant role in promoting animal growth, improving feed utilization efficiency, and reducing diseases. Figure 5 shows that the Chao-1 and Shannon indexes of CM and HCM treatments were significantly different (*p* < 0.05) from that of the control after 12 h of fermentation, revealing the induction of richness and the variety of gut microbiota. However, their Simpson indices were lower than that of the control. The addition of CM and HCM may have promoted the growth of the least prevalent bacterial group. This is not consistent with the impact of xylo-oligosaccharides [26] and isomalto-oligosaccharides [27] on weaned pigs. This could be attributed to the different types of oligosaccharides used, as the effects of different types of functional oligosaccharides in animals are not the same [23].

To better understand the specific changes in the microbial community, the taxonomic composition of different groups was examined (Figure 6). At the phylum level, *Firmicutes*, *Proteobacteria*, *Actinobacteria*, and *Bacteroidetes* were the top four abundant phyla in the different treatments. No significant difference (*p* > 0.05) among bacterial groups was observed after 12 h of fermentation. *Actinobacterota* and *Firmicutes* abundances were higher, but the *Proteobacteria* load was lower with CM treatment compared to the control treatment, which was not significantly different (*p* > 0.05). In the HCM treatment, *Actinobacterota* and *Proteobacteria* abundances were higher (*p* > 0.05), while the *Firmicutes* abundance was lower compared with the control treatment. It has been documented that the most abundant phylum-level microbiota across the weaning piglet gastrointestinal tract are *Firmicutes*, *Proteobacteria*, and *Bacteroidetes* [28]. *Actinobacteria* is an effective probiotic due to its capacity for antimicrobial action, fish immune response, growth promotion, gut condition resistance, and water quality improvement, among others [29]. The high *Proteobacteria* abundance in HCM may have been fueled by a high concentration of monosaccharides as revealed in the hydrolyzed copra meal section above. *Proteobacteria* include a wide variety of pathogens but are thought to be important in preparing the gut for colonization by the strict anaerobes required for healthy gut function by consuming oxygen and lowering the redox potential in the gut environment [30]. *Firmicutes* are highly abundant in the pig GI tract and are involved in maintaining energy balance in the body [31]. The findings suggest that the CM and HCM may stimulate the composition of the gut microbiota, but more research with higher sample concentrations is needed.

At the family level, *Erysipelotrichaceae*, *Ruminococcaceae*, *Lactobacillaceae*, *Clostridiaceae*, and *Lachnospiraceae* abundances were predominant (Figure 7). The other bacterial groups identified include *Peptostreptococcaceae*, *Enterococcaceae*, *Planococcaceae*, Enterobacteriaceae, and *Methanobacteriaceae*. Compared with the control treatment, the relative abundance of these bacteria under CM and HCM treatments was not significantly different (*p* > 0.05) after fermentation. After 12 h of fermentation, Clostridiaceae_1, *Erysipelotrichaceae*, and Enterobacteriaceae *Lachnospiraceae*, *Lactobacillaceae*, and *Ruminococcaceae* decreased in the control treatment. Similarly, the abundance of these bacteria was reduced in the CM and HCM treatments, but the relative abundance of *Lachnospiraceae*, *Peptostreptococcaceae*, *Ruminococcaceae*, and *Lactobacillaceae* was higher in the CM and HCM compared to the control. In comparison to the control, the load of Enterobacteriaceae was lower in CM but higher in HCM, whereas *Erysipelotrichaceae* declined in both CM and HCM treatments. The results showed that CM and HCM supplementation may aid in the long-term survival of *Lactobacillaceae*, *Peptostreptococcaceae*, *Lachnospiraceae*, and *Ruminococcaceae* populations when compared to the control at 12 h. Some researchers discovered that swine fed a high-fiber diet with arabinoxylan–oligosaccharide (AXOS) supplements had more *Lachnospiraceae* in the ileal lumen [32]. The growth and health of pigs were promoted by combining probiotics and soybean oligosaccharides (SBOS), which increased *Ruminococcaceae* and other beneficial bacteria [33].

#### 3.3.2. Short-Chain Fatty Acid during Fermentation

As shown in Figure 8, Figure 9, Figure 10 and Figure 11, the production of various SCFAs during swine microbiota fermentation in the presence of CM and HCM was not significantly different (*p* > 0.05) when compared with the control. As can be seen from the microbiological data (Figure 7), CM and HCM did not exert a significant effect on the growth of *Enterobacteriaceae* or lactic acid bacteria (no statistically significant difference was found between the CM and HCM and the control). The SCFAs observed were acetic acid, propionic acid, lactic acid, and butyric acid. Some metabolites such as acetic acid (Figure 8) and butyric acid (Figure 11) appeared to increase from 0 h to 12 h of fermentation but were not significantly different (*p* > 0.05), showing that CM and HCM fermentation could positively influence acetic and butyric acids in swine microbiota. Propionic acid (Figure 10) was reduced while lactic acid (Figure 9) was seen to be maintained after the fermentation of CM and HCM. Acetic acid has a role in lowering abdominal fat accumulation and protecting the liver from lipid accumulation [34]. Butyric acid, the most prominent SCFA, is the primary fuel source for colon cells and contributes to colorectal cancer prevention, inflammation inhibition, thermogenesis stimulation, and resistance to obesity [35]. Propionic acid is highly effective at promoting weight loss, reducing inflammation and cholesterol, protecting against diet-induced obesity, and regulating gut hormones [36]. According to the findings, CM and HCM may not have a noticeable effect on the SCFA richness in swine microbiota but could maintain their concentrations.

Gut microbiota is a large and complex microecosystem located in the intestine, and many polysaccharides have been certified to have the ability to regulate gut microbiota composition and structure [37]. The impact of HCM on human gut microbiota has been indicated [18]. However, the health effects of HCM in modulating the microbiota are lacking, especially in swine. To address this, the stimulation effects of HCM in the fecal microbiota of swine were evaluated. Previous work indicated that CM and HCM were digested during gastric and intestinal digestion, and HCM showed a higher hydrolysis degree with more oligosaccharide release than CM. The high hydrolysis level in HCM was attributed to pre-hydrolysis with beta-mannanase, which breaks down complex polysaccharides to eliminate antinutritional factors that impair digestion. This could ensure an improved energy and nutrient utilization performance [5], and a further understanding of HCM in modulating the gut microbiota will propel its application. In this study, HCM had a positive effect on bacteria richness and diversity over time, which might encourage the growth of the bacterial group that is least prevalent. It was recorded that the intake of copra meal hydrolysate at 3 g/d led to increased microbial diversity and richness in healthy adults [18]. HCM did not significantly improve bacterial groups, though the levels of *Proteobacteria* and *Actinobacterota* were higher compared with CM and control treatments. Also, HCM supplementation could maintain the level of *Erysipelotrichaceae*, *Lachnospiraceae*, and *Ruminococcaceae*. *Lachnospiraceae* benefit the host by producing short-chain fatty acids for immune stimulation—and intestinal acidification [38]. *Ruminococcaceae* is an important butyrate producer and bacterium in intestinal health [39]. In addition, HCM may have a positive influence on acetic acid and propionic acid richness in swine microbiota. Acetic acid and propionic acid are the two major SCFAs responsible for anti-inflammatory, immunoregulatory, anti-obesity, anti-diabetes, cardiovascular protective, hepatoprotective, and neuroprotective activities in the gut microbiota [35].

Copra meal has numerous advantages when used as animal feed. The ability to overcome the limitations of CM and to effectively utilize it in the gastrointestinal tract of swine is of great interest. There is a wide variation found in the composition of carbohydrates in different feeds and feed ingredients, which has a significant impact on their physicochemical properties and digestion and fermentation characteristics in the gastrointestinal tract of animals. For example, it has been reported that copra meal and defatted copra meal comprised protein contents of 4.7 and 19.2% (*w*/*w*), carbohydrate values of 25.3 and 80.1% (*w*/*w*), fat contents of 72.1 and 0% (*w*/*w*), ash contents of 1.8 and 1.6% (*w*/*w*), and moisture contents of 2.7 and 2.8% (*w*/*w*), respectively [40,41]. However, the high fiber content tends to interfere with the digestibility of copra meal, resulting in the retention of nitrogen and other nutrients. This is an essential factor that limits the use of copra meal in simple-stomached animals, including swine.

Previous research has found that the high fiber content of copra meal results in lower digestible and metabolizable energy concentrations than corn [42]. Also, the protein quality in copra meal is less than those of soybean meal and palm kernel products in weanling swine [43]. Thus, dietary fiber is not only indigestible to the digestive enzymes of mammals, but it can also reduce the digestibility of nutrients and the efficiency of energy utilization [44]. Swine do not degrade these fibers, because swine cannot excrete non-starch polysaccharide (NSP) enzymes. However, mannanases from *Bacillus circulans* NT 6.7 are capable of breaking down dietary fiber and releasing significant nutrients within the fiber fraction of the consuming animal. The NSP of copra meal is in mannan, galactomannan, and cellulose (26, 61, and 13%, respectively). Therefore, in this study, mannanase treatment that yielded HCM has an improved nutrient digestibility and enables efficient utilization in swine based on three beneficial properties. The following can be deduced: firstly, the breakdown of polysaccharides eliminates the antinutritional factors present in copra meal that impair the performance of swine. Secondly, the degraded carbohydrates and proteins readily absorbed can supply the nutrients needed by the swine for maintenance and production [5]. Finally, the mannooligosaccharides produced can function as prebiotics to improve the swine’s health. According to the current study, although the effect of HCM on the swine gut microbiota is similar to that of CM based on the above results, HCM is a better alternative due to the aforementioned beneficial properties (nutrient availability and efficient utilization).

## 4. Conclusions

Hydrolyzed copra meal (HCM) produced by mannanase from *Bacillus circulans* NT 6.7 consisted of 25.74 ± 0.13 mg/mL of reducing sugar and 30.95 mg/mL of total oligosaccharide. SEM images confirmed the breakdown of bio-compounds in HCM to release nutrients and reduce antinutritional and fiber contents. The supplementation of CM and HCM could not significantly modify the gut microbiota of swine by increasing bacteria richness and diversity at different times. However, CM and HCM could maintain a substantial abundance of *Erysipelotrichaceae, Lachnospiraceae,* and *Ruminococcaceae* in the long run when compared to the control. HCM could preserve acetic acid and butyric acid richness in swine microbiota. HCM may be a more efficient alternative feed for swine than CM in terms of high digestibility, antinutritional factor removal, and impact on the gut microbiota. However, more research is needed to determine the effect of different HCM concentrations on swine gut microbiota.

## Figures and Tables

**Figure 1 animals-14-01677-f001:**
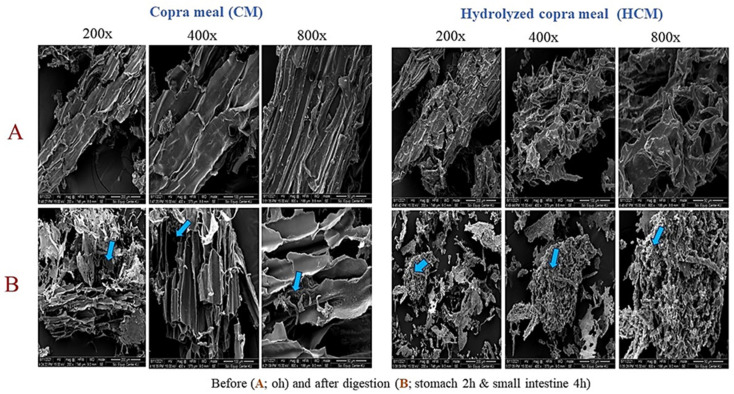
Scanning electronic microscope (SEM) images of swine simulated GI digested copra meal (CM) and hydrolyzed copra meal (HCM) at different magnifications. HCM displayed more obvious breakages of connective structures with surface holes compared to CM. HCM showed a higher polymerization of oligosaccharides. Blue arrows indicate holes and breakages resulting from hydrolysis.

**Figure 2 animals-14-01677-f002:**
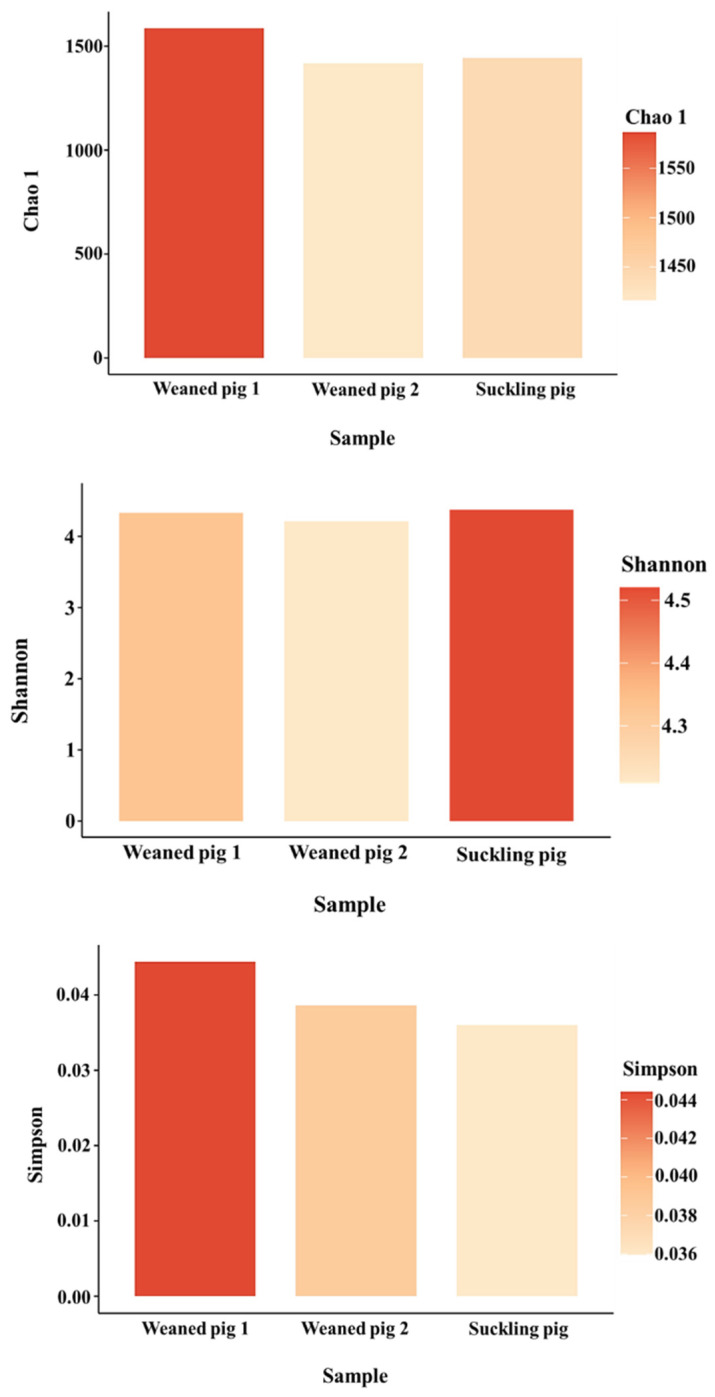
Alpha diversity of the microbiota of fresh feces in the suckling pig and weaned pigs. Bacterial richness and diversity seemed to differ among the examined pigs, whereas the variation in abundance appeared to be similar.

**Figure 3 animals-14-01677-f003:**
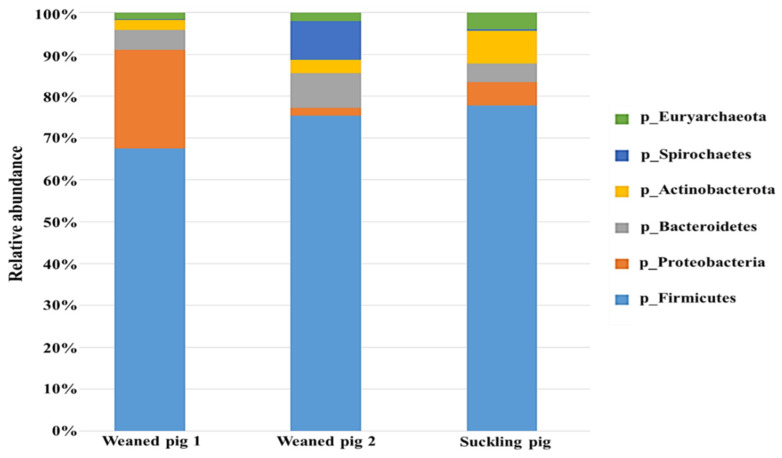
Comparison of operational taxonomic units (OTUs) and classification abundance of bacteria in fresh feces of suckling pig and weaned pigs. The relative abundance of the dominant bacteria (percentages) of swine at the phylum level. Data are expressed as means, *n =* 2. No significant variation (*p* > 0.05) in the abundance was observed among the three swine.

**Figure 4 animals-14-01677-f004:**
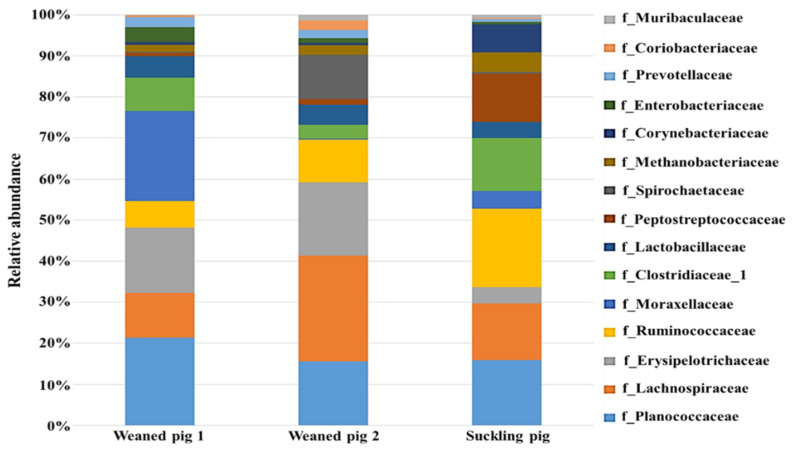
Comparison of operational taxonomic units (OTUs) and bacterial abundance classification in suckling and weaned pig feces. The relative abundance of the dominant bacteria in swine (in percentages) at the family level. Data are expressed as means, *n =* 2. No significant variation (*p* > 0.05) in the abundance was observed among the three swine.

**Figure 5 animals-14-01677-f005:**
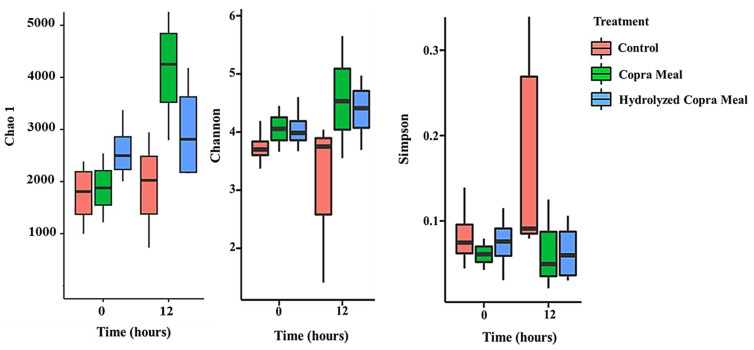
Alpha diversity of the microbiota of control, copra meal, and hydrolyzed copra meal in the feces of swine, examined in an in vitro vessel. Significantly higher indexes were identified at *p* < 0.05. CM and HCM may have promoted the growth of the least prevalent bacterial group.

**Figure 6 animals-14-01677-f006:**
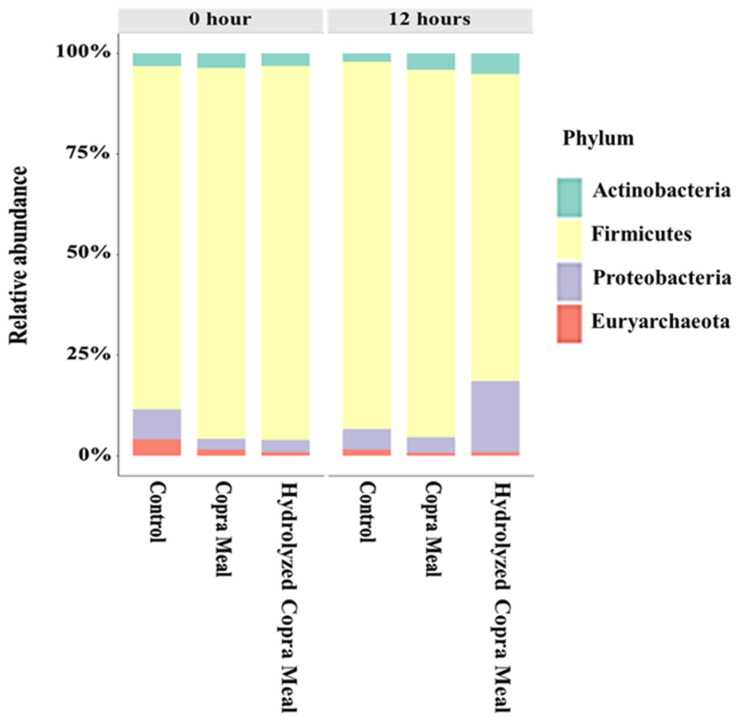
Alterations in bacterial operational taxonomic units (OTUs) and classification abundance at the phylum level. The relative abundance of the dominant bacteria (percentages) of weaned swine (*n =* 2). Control; copra meal, CM; hydrolyzed copra meal, HCM. No significant variation was observed in phylum level among treatments.

**Figure 7 animals-14-01677-f007:**
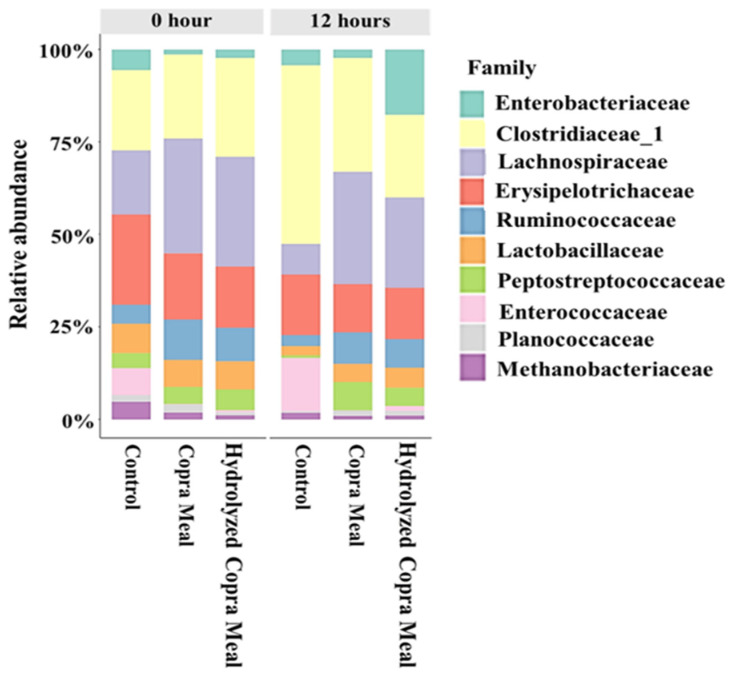
Alterations in bacterial operational taxonomic units (OTUs) and classification abundance at the family level. The relative abundance of the dominant bacteria (percentages) of weaned swine (*n =* 2). Control; copra meal, CM; hydrolyzed copra meal, HCM. Both CM and HCM maintained the population of *Lactobacillaceae*, *Peptostreptococcaceae*, *Lachnospiraceae*, and *Ruminococcaceae*.

**Figure 8 animals-14-01677-f008:**
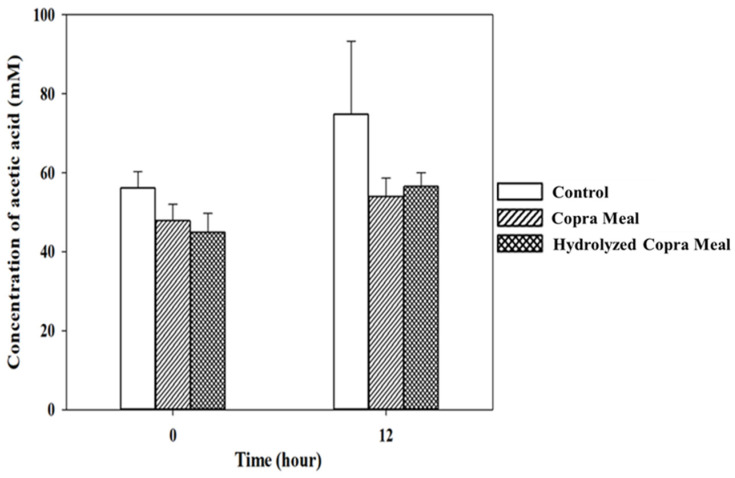
The concentrations of acetic acid in swine feces gut fermentation. Data are expressed as means ± SD, (*n =* 2). Control; copra meal, CM; hydrolyzed copra meal, HCM. CM and HCM significantly lowered the content of acetic acid in swine microbiota.

**Figure 9 animals-14-01677-f009:**
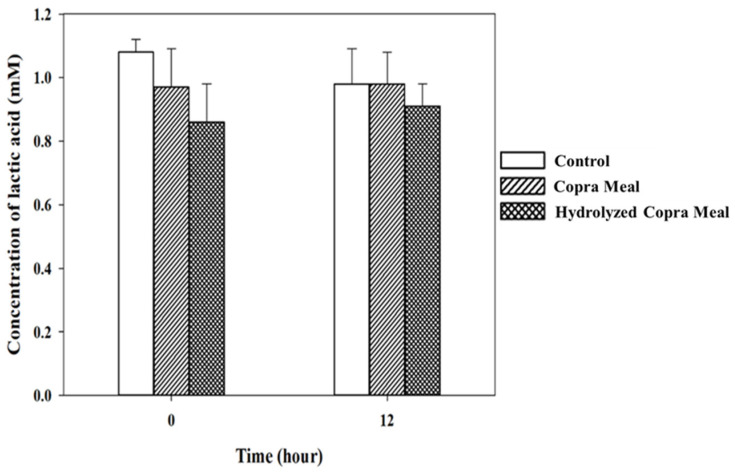
The concentrations of lactic acid in swine feces gut fermentation. Data are expressed as means ± SD, (*n =* 2). Control; copra meal, CM; hydrolyzed copra meal, HCM. CM and HCM may not have a notable effect on the diversity of lactic acid in swine microbiota but could maintain its levels.

**Figure 10 animals-14-01677-f010:**
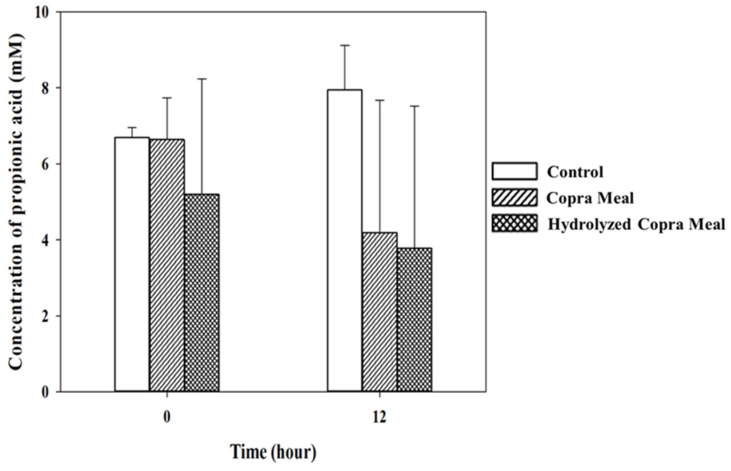
The concentrations of propionic acid in swine feces gut fermentation. Data are expressed as means ± SD (*n =* 2). Control; copra meal, CM; hydrolyzed copra meal, HCM. CM and HCM might not significantly reduce the richness of propionic acid in swine microbiota.

**Figure 11 animals-14-01677-f011:**
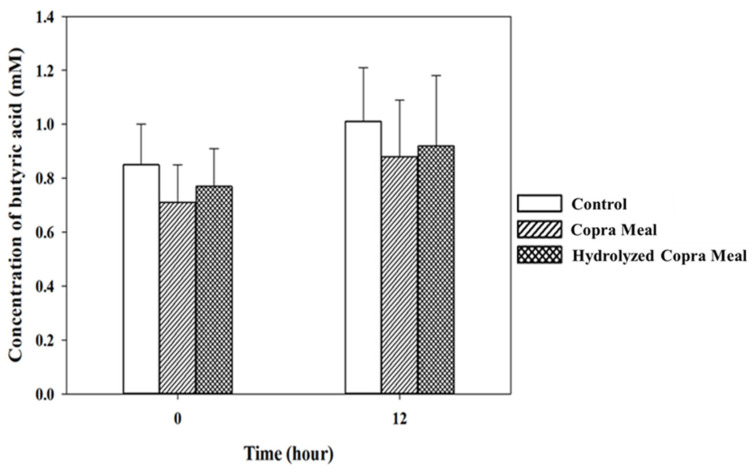
The concentrations of butyric acid in swine feces gut fermentation. Data are expressed as means ± SD, (*n =* 2). Control; copra meal, CM; hydrolyzed copra meal, HCM. CM and HCM might not significantly impact the richness of butyric acid in swine microbiota but could preserve its concentrations.

**Table 1 animals-14-01677-t001:** The amount of copra meal and copra meal hydrolysate during digestion in dry weights (g).

Simulated Gastrointestinal (GI) Conditions	Copra Meal (CM)	Hydrolyzed Copra Meal (HCM)
Before digestion (0 h)	1.0	1.0
Stomach (2 h)	0.94	0.70
Small intestine (4 h)	0.82	0.68

**Table 2 animals-14-01677-t002:** The content of oligosaccharide (mg, dry weights) released during digestion in swine simulated conditions.

	Copra Meal (CM)					Hydrolyzed Copra Meal (HCM)				
	M6	M5	M4	M3	M2	M6	M5	M4	M3	M2
Stomach	275.44					278.92	3.64	11.00	16.44	6.88
Small intestine	405.72	0.54		9.30				2.40	3.36	0.060

## Data Availability

Data are contained within the article.

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
