# Peer review of "Simulated Swine Digestion and Gut Microbiota Fermentation of Hydrolyzed Copra Meal"

_animals, 2024, doi:10.3390/ani14111677_

Round 1
Reviewer 1 Report
Comments and Suggestions for Authors
Introduction- I suggest expanding the introduction with more information about MOS health benefits and their impact on modulation of the gut microbiota
56 I suggest change from “According to [8] copra […]” to “According to Prayoonthien et al., 2018 copra […]KUB-E010 growth [8,9].”
115 Please, rewrite „ 2.5 Feces collection and analysis of intestinal microbiota structure” in more understandable English
How did the authors ensure that the samples were not additionally contaminated (i.e by the collector) from the collection process to the inoculation stage?
NGS sequencing/library preparation/bioinformatics – was it outsourced or done directly by the authors ?
References – many references used by authors are older than 5 year old, if possible please use more up-to-date references
From my point of view I would like to suggest two ideas for the authors for the further research.
a) trying some kind of symbiotic preparation
b) testing how HCM affects wanted and unwanted microbiota
115 Please, rewrite „ 2.5 Feces collection and analysis of intestinal microbiota structure” in more understandable English
Author Response
We already revised our manuscript accordingly in the PDF file attached

Reviewer 2 Report
Comments and Suggestions for Authors
Dear Authors,
Strictly follow my comments/corrections in the attached file, my general comments are as follows:
1. In the introduction section, the need for the study should be ideally raised followed by the hypothesis of the study.
2. In statistical analysis, add a mathematical model for a better understanding of data analysis.
3. In the results and discussion section, add logical reasoning for each result before discussing with the previous studies.
4. In the conclusion section, add the limitations and implications of the study.
Thank You!

Author Response

(The authors gave the same response as above.)

Round 2
Reviewer 2 Report
Comments and Suggestions for Authors
Good manuscript.